# Association between the severity of hard-to-treat psoriasis and the prevalence of metabolic syndrome: A hospital-based cross-sectional study in Jakarta, Indonesia

Dina Evyana[1], Endi Novianto[1]*, Windy K. Budianti[1], Roro I. A. Krisanti[1], Wismandari Wisnu[2], Retno Wibawanti[3], Hanny Nilasari[1], Lili Legiawati[1], Saskia A. R. Hapsari[4], Euis Mutmainnah[5]

1 Faculty of Medicine Universitas Indonesia, Department of Dermatology and Venereology, Dr. Cipto Mangunkusumo National Central General Hospital, Jakarta, Indonesia, 2 Faculty of Medicine Universitas Indonesia, Department of Internal Medicine, Dr. Cipto Mangunkusumo National Central General Hospital, Jakarta, Indonesia, 3 Faculty of Medicine Universitas Indonesia, Department of Community medicine, Dr. Cipto Mangunkusumo National Central General Hospital, Jakarta, Indonesia, 4 Department of Dermatology and Venereology, Tarakan General Hospital, Jakarta, Indonesia, 5 Department of Dermatology and Venereology, Persahabatan Central General Hospital, Jakarta, Indonesia

* endinoviantodr@gmail.com

## Abstract

Psoriatic lesions on the scalp, face, intertriginous, genitals, palms/soles, and nails are often delay diagnosed, hard-to-treat, and cause disability. Metabolic syndrome (MetS) is one of the most frequent and significant comorbidities in psoriasis. Many studies have discovered a link between psoriasis and MetS, but none have specifically assessed the hard-to-treat psoriasis in Indonesian population. This is a multicenter study involving four dermatology referral hospitals to investigate the association between psoriasis severity that has hard-to-treat lesions with the prevalence of MetS in Jakarta, Indonesia. Data was collected from April to October 2022. The severity of 84 hard-to-treat psoriasis patients was measured by Psoriasis Area Severity Index (PASI) scores. The participants divided into PASI score >10 (severe) and ≤ 10 (mild-moderate) groups. MetS was identified based on the modified National Cholesterol Education Program Adult Treatment Panel III. MetS was found in 64.3% of patients. Patients with a PASI score>10 had a significantly higher risk of metabolic syndrome compared to those with a score ≤ 10 (78.6% vs 50%, OR 3.667; 95% CI 1.413–9.514; $p$ = 0.006). The prevalence of hypertension ($p$ = 0.028), low levels of high-density lipoprotein (HDL) cholesterol ($p$ = 0.01), mean fasting blood sugar ($p$ = 0.018), and triglyceride levels ($p$ = 0.044) between the two groups differed significantly. This study found most frequent components of MetS were abdominal obesity, decreased levels of HDL cholesterol, hypertension, hyperglycemia, and hypertriglyceridemia respectively. Individuals with severe hard-to-treat psoriasis had a 3.67 times more likely to have MetS rather than the mild-moderate group.

**Data Availability Statement:** All relevant data are within the manuscript and its Supporting information files.

**Funding:** This study was supported by International Indexed Publication (Publikasi Terindeks Internasional/PUTI) 2022 research grant from the Directorate of Research and Development Universitas Indonesia (No. NKB-155/UN2.RST/HKP.05.00/2022). The funders had no role in study design, data collection and analysis, decision to publish, or preparation of the manuscript.

**Competing interests:** The authors have declared that no competing interests exist.

# Introduction

Psoriasis is a chronic, immune-mediated, systemic inflammatory disorder that is associated with multiple comorbidities, and MetS is one of the most frequent and significant comorbidities [1–3]. The existence of MetS in psoriasis patients has been proven in numerous research [2,4,5]. The global prevalence of MetS in adult psoriasis was reported 32% [6]. The prevalence of MetS was 39% in the Indonesian general population [7]. Both in adult and pediatric populations, studies have also demonstrated that elements of the MetS (abdominal obesity, hypertension, insulin resistance, dyslipidemia) were more prevalent in individuals with psoriasis than in those without it [1,5]. The occurrence of MetS varies depending on the severity of the disease [2]. Severe psoriasis patients were found to have the highest risk of cardiometabolic disorders [5]. MetS directly elevates the risk of CVD and premature mortality in psoriasis patients, significantly lowering their expected lifespan [1].

The clinical manifestations of psoriasis range from scaly erythematous plaques limited to the elbows, knees, or scalp, to severe conditions affecting the entire skin surfaces [8]. Involvement of the scalp, face, intertriginous, genitals, palms, soles, and nails in psoriasis are often delay diagnosed, hard to treat, and cause disability [9–11]. Those predilections were known as *hard-to-treat* (HTT), *difficult-to-treat*, or *challenging-to-treat* areas [10–12]. According to the Danish Skin Cohort 2020 around 64.8% of 4016 adult plaque psoriasis patients had lesions in at least one HTT area [10]. This finding was consistent with the Corona Psoriasis Registry 2018 which revealed two-thirds of 2042 patients had at least one HTT area [12]. Despite the limited surface area frequently impacted by psoriasis within those regions, it has major physical, and psychosocial impacts, along with the risk of comorbidities [10,11,13].

The pathogenesis of MetS in psoriasis remains elusive; however, shared inflammatory signaling, molecular mediators, hereditary vulnerability, and similar factors associated are suspected to play a role [5,14]. Psoriasis and MetS are characterized by persistent proinflammatory conditions [1,15]. Dysregulated levels of adipokines, cytokines (such as TNF-α, VEGF, IL-6, IL-8, and IL-12), and inflammatory immune response, driven by Th1, Th19, and dysfunctional adipose tissue; were shown to be responsible for the progression of psoriasis and MetS. Moreover, a study has demonstrated that the migration of proinflammatory mediators into the systemic circulation worsens the severity of psoriasis and MetS by impairing vascular endothelial function and increasing oxidative stress [15]. Since MetS and psoriasis develop from both genetic and environmental factors, the relationship between psoriasis and MetS may differ amongst different ethnicities or populations due to differences in environmental exposure and genetic background [13,16]. Although most studies have shown link between psoriasis severity and MetS, similar research among Asian psoriatic patients were limited with varying results [13,17]. It is not always feasible to generalize the data and findings beyond populations [13]. Therefore, the aim of the present study was to determine whether hard-to-treat psoriasis severity is associated with the increased prevalence of MetS and which components of MetS play an important role in our Indonesian HTT psoriasis population.

# Materials and methods

## Study design

This cross-sectional study included adult plaque psoriasis patients with involvement in at least one hard-to-treat area, 18 years and older, recruited consecutively from the dermatology & venereology clinics at 3 academic hospitals and 1 private hospital in Jakarta, Indonesia from April to October 2022. The diagnosis of hard-to-treat psoriasis was made by a dermatologist. Exclusion criteria were pregnancy, having other diseases that interfere with measuring waist

circumference (such as intra-abdominal tumors), onychomycoses, and reduction of Psoriasis Area Severity Index (PASI) score of more than 10% after receiving psoriasis therapy compared to the initial score before treatment. Approval for this study was obtain from each hospital according to the principles of the Declaration of Helsinki, include the Health Research Ethics Committee—Faculty of Medicine Universitas Indonesia/Cipto Mangunkusumo Central General Hospital (HREC-FMUI/CMH) (KET-231/UN2.F1/ETIK/PPM.00.02/2022), HREC—Tarakan General Hospital (005/KEPK/RSUDT/2022), HREC—Persahabatan Central General Hospital (24/KEPK-RSUPP/03/2022) and Sam Marie Hospital Jakarta (include in the amendments of research protocol from HREC-FMUI/CMH No. ND-273/UN2.F1/ETIK/PPM.00.02/2022).

## Data collection

Data were acquired through patient assessments, physical examinations, and laboratory investigations. Informed written consent were obtained from all subjects enrolled in the study. Only the first and second authors had access to information that could identify individual participants during or after data collection. The interview was conducted to obtain details about socio-demographic backgrounds and risk factors such as age, gender, educational level, smoking habits, alcohol intake, onset, duration, family history, treatment of psoriasis, and pre-existing MetS. The severity of psoriasis was assessed using the PASI score through physical examinations. Psoriasis was classified as severe if the PASI score was >10 and mild to moderate if its score was ≤10 [2,13]. Using a flexible tape OneMed® OD-235, the waist circumference was measured midway between the iliac crest and the lowest rib [7]. Body weight and height was measured using the Tesena® TSN 9806 WHS medical digital weighing scale and height measuring. The weight measurement is recorded in kilograms (kg) and height in meters (m). Body weight (kg) divided by height squared ($m^2$) to calculate body mass index (BMI). Blood pressure was measured using Ommron® HBP-1300 blood pressure monitors.

Following an overnight fast, subjects' venous blood samples were taken and evaluated for triglycerides, HDL cholesterol, glucose, and HbA1c at the time of physical examination. Metabolic syndrome was defined if three or more of the following five criteria are met according to the modified National Cholesterol Education Program Adult Treatment Panel (NCEP ATP) III: (i) abdominal obesity, defined by waist circumference ≥90 cm in men or ≥80 cm in women; (ii) blood pressure ≥130/85 or treatment for hypertension; (iii) serum triglyceride level ≥150 mg/dl or treatment for elevated triglycerides; (iv) serum HDL level <40 mg/dl in men or <50 mg/dl in women or treatment for low HDL; and (v) fasting blood sugar ≥100 mg/dl or treatment for hyperglycemia [18]. The laboratory tests were conducted at the Clinical Pathology Laboratory FMUI/CMH using the Abbott Architect ci8000 (Germany).

## Statistical analysis

Statistical analyses were made using IBM SPSS Statistics (ver. 20, USA) software program. Except noted otherwise, continuous variables were presented as means ± SD. The categorical variables were presented as numbers and percentages. Data were considered statistically significant if the $p$-value < 0.05. To evaluate the association between two categorical variables, the Chi-squared test was performed. The mean values of continuous variables including age, body mass index, disease duration, age at onset, PASI score, and laboratory parameter levels either between HTT psoriasis patients with and without MetS or between PASI score<10 and >10 were compared by an independent t-test. Flow diagram of the study population is shown in Fig 1.

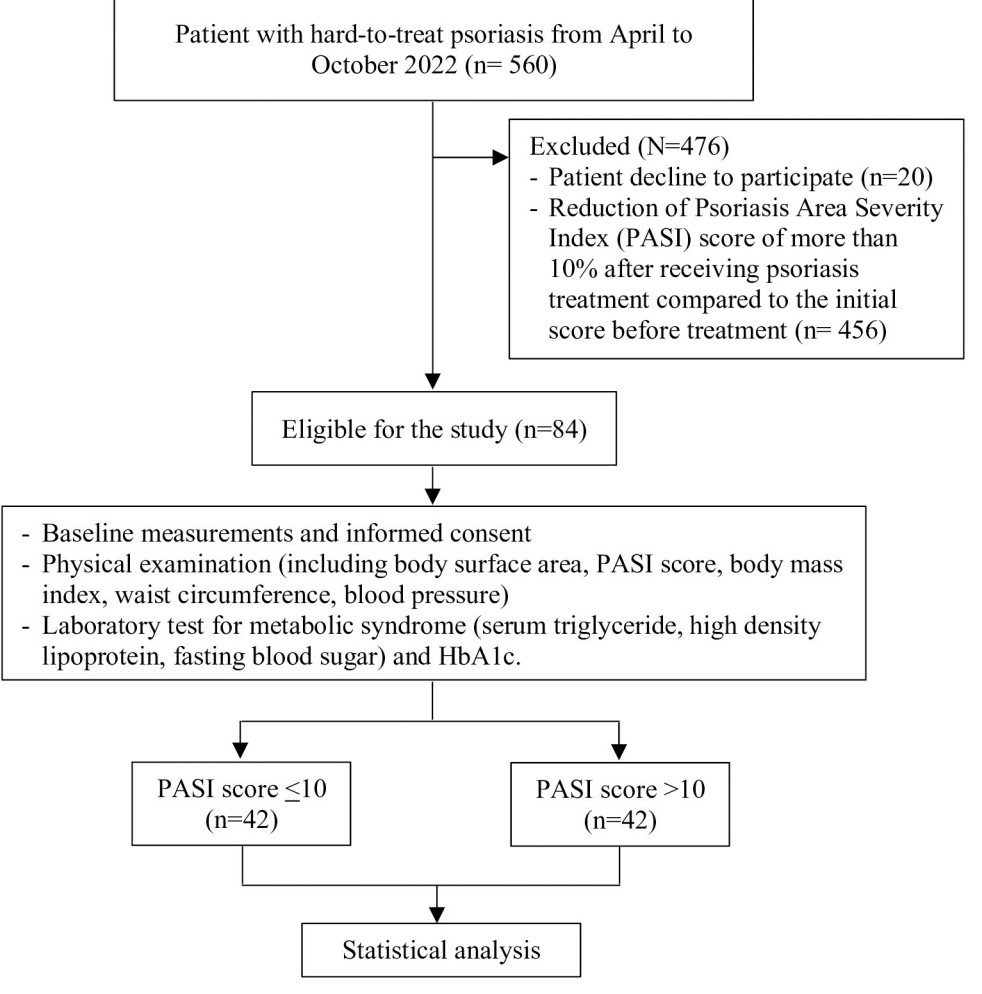

**Fig 1. Flow diagram of the study population.**

## Results

### Sociodemographic characteristics of the study population

The participants (n = 84) were divided into hard-to-treat psoriasis with MetS (n = 54) and without MetS (Non-MetS, n = 30). Afterwards the participants were also divided into PASI ≤10 (mild = 12, moderate = 30, total = 42) and PASI>10 groups (severe = 42). There were no missing data in this study. Tables 1 and 2 shows the baseline features of HTT psoriasis subjects with and without MetS. Patients with MetS were older (mean age 49.15 ±11.87 years) with a slightly higher proportion of females both in the MetS group (55.6%) and over-all (51.2%). Most of the subjects have high educational levels, are employed and covered by the national health insurance system. Smoking (20.2%) and alcohol (7.1%) habits were found to be less in all groups. The majority of subjects had abdominal obesity (84.5%) and were obese grade I (47.6%), average waist circumference was 99.85±10.37 cm and BMI 28.77±4.83 kg/m$^2$ in the MetS group compared to the other group. Those are consistent with the BSA and PASI scores which were also found to be higher in the MetS group. Based on the BSA, the subjects were dominated by severe psoriasis, as well as according to the PASI score. Most

**Table 1. Sociodemographic characteristic of the study population.**

| Variable | Total (*n* = 84) | MetS (*n* = 54) | Non-MetS (*n* = 30) | *p*-value |
|---|---|---|---|---|
| Age, years^ | 44.13 ±13.91 | 49.15 ±11.87 | 35.1±12.84 | <0.000> |
| Sex, n (%) | | | | 0.283< |
| Male | 41 (48.8) | 24 (44.4) | 17 (56.7) | |
| Female | 43 (51.2) | 30 (55.6) | 13 (43.3) | |
| Educational level, n (%) | | | | 0.3+ |
| Low | 4 (4.8) | 4 (7.4) | 0 (0) | |
| Middle | 42 (50) | 28 (51.9) | 0 (0) | |
| High | 38 (45.2) | 22 (40.7) | 30 (100) | |
| Employment status, n (%) | | | | 0.794< |
| Yes | 46 (54.8) | 29 (53.7) | 17 (56.7) | |
| No | 38 (45.2) | 25 (46.3) | 13 (43.3) | |
| Financing status, n (%) | | | | 0.180+ |
| National health insurance | 78 (92.9) | 52 (96.3) | 26 (86.7) | |
| Other health insurance | 0 (0) | 0 (0) | 0 (0) | |
| Private | 6 (7.1) | 2 (3.7) | 4 (13.3) | |
| Smoking status, n (%): | | | | 0.599< |
| Current smokers | 17 (20.2) | 10 (18.5) | 7 (23.3) | |
| Non-smokers | 67 (79.8) | 44 (81.5) | 23 (76.7) | |
| Alcohol consumption, n (%) | | | | 0.662+ |
| Drinkers | 6 (7.1) | 3 (5.6) | 3 (10) | |
| Non-drinkers | 78 (92.9) | 51 (94.4) | 27 (90) | |

^Data in mean ± standard deviation;

#Data in median & interquartile range;

<Chi Square,

>T-independent;

*Mann-Whitney;

+Fisher's exact;

MetS: Metabolic syndrome; BMI: Body mass index; BSA: Body surface area; PASI: Psoriasis area severity index; FBG: Fasting blood glucose; TG: Triglycerides; HDL: High density lipoporotein; M: Male; F: Female.

of the subjects did not have a family history of psoriasis (15%) and had an early-onset disease (75%).

Almost all subjects experienced scalp psoriasis up to 96.4%, followed by folds, facial, nails, genital, and palmoplantar. This research found most treatment of psoriasis in combination topical and systemic (79.8%). The most widely used systemic treatment was methotrexate (61.9%) followed by phototherapy and cyclosporine. More than half of the patients had hypertension with greater proportions in the MetS group (23.3% vs 72.2%, *p*<0.000). The proportion of diabetes and prediabetes according to HbA1c level dominates the MetS group. There were differences found in the level of HbA1c and proportions of the HbA1c category between MetS and Non-MetS. The components of MetS in the whole subjects and MetS group were similar and the highest proportion was obesity (84.5%; 98.1%), followed by low HDL level (67.9%; 90.7%), hypertension (54.8%; 72.2%), hypertriglyceridemia (41.7%; 62.9%), and hyperglycemia (30.9%; 46.3%) respectively. All the components of MetS were different in both groups and statistically significant as shown in Table 2.

**Table 2. Baseline clinical characteristic of the study population.**

| Variable | Total (n = 84) | MetS (n = 54) | Non-MetS (n = 30) | p-value |
|---|---|---|---|---|
| Waist circumference, cm^ | 94.98±13.43 | 99.85±10.37 | 86.2±13.99 | <0.000> |
| Abdominal obesity, n (%) | 71 (84.5) | 53 (98.1) | 18 (60) | <0.000< |
| BMI, kg/m²^ | 27.25±5.26 | 28.77±4.83 | 24.52±4.95 | <0.000> |
| BMI level, n (%) | | | | <0.000+ |
| Underweight (<18.5 kg/m²) | 1 (1.2) | 0 (0) | 1 (3.3) | |
| Normal (18.5–22.9 kg/m²) | 17 (20.3) | 4 (7.4) | 13 (43.3) | |
| Overweight (23–24.9 kg/m²) | 8 (9.5) | 5 (9.3) | 3 (10) | |
| Obese I (25–29.9 kg/m²) | 40 (47.6) | 30 (55.5) | 10 (33.3) | |
| Obese II (≥30 kg/m²) | 18 (21.4) | 15 (27.8) | 3 (10) | |
| Hypertension, n (%) | 46 (54.8) | 39 (72.2) | 7 (23.3) | <0.000< |
| Family history, n (%) | 13 (15.5) | 6 (11.1) | 7 (23.3) | 0.207+ |
| Onset, n (%) | | | | 0.018< |
| ≥ 40 years (early onset) | 63 (75) | 36 (66.7) | 27 (90) | |
| >40 years (late onset) | 21 (25) | 18 (33.3) | 3 (10) | |
| Duration (years)#^ | 10 (4.25–18) | 13.53±10.31 | 8 (4–14) | 0.132* |
| PASI# | 10.1 (6.35–16.72) | 10.95 (6.85–16.37) | 8.6 (6.07–18.17) | 0.265* |
| *Severity in PASI*, n (%) | | | | 0.024+ |
| Mild: < 5 | 12 (14.3) | 6 (11.1) | 6 (20) | |
| Moderate: 5–10 | 30 (35.7) | 15 (27.8) | 15 (50) | |
| Severe: > 10 | 42 (50) | 33 (61.1) | 9 (30) | |
| *Hard-to-treat areas*, n (%) | | | | |
| Scalp | 81 (96.4) | 53 (98.1) | 28 (93.3) | 0.289+ |
| Face | 61 (72.6) | 37 (68.5) | 24 (80) | 0.258< |
| Inverse | 64 (76.2) | 42 (77.8) | 22 (73.3) | 0.647< |
| Genital | 24 (28.6) | 13 (24.1) | 11 (36.7) | 0.221< |
| Nails | 54 (64.3) | 37 (68.5) | 17 (56.7) | 0.277< |
| Palmoplantar | 20 (23.8) | 11 (20.4) | 9 (30) | 0.321< |
| Psoriasis treatment, n (%) | | | | 0.082< |
| Topical only | 17 (20.2) | 14 (25.9) | 3 (10) | |
| Systemic & topical | 67 (79.8) | 40 (74.1) | 27 (90) | |
| Systemic treatment, n (%) | | | | |
| Phototherapy | 24 (28.6) | 13 (24.1) | 11 (36.7) | 0.126< |
| Methotreaxate | 52 (61.9) | 33 (61.1) | 19 (63.3) | 0.841< |
| Cyclosporine | 9 (10.7) | 3 (5.5) | 6 (20) | 0.333+ |
| Acitretin | 1 (1.2) | 0 (0) | 1 (3.3) | 0.125+ |
| Biologic | 2 (2.4) | 1 (1.9) | 1 (3.3) | 0.670< |
| Steroid | 3 (3.6) | 3 (5.5) | 0 (0) | 0.535+ |
| Others | 6 (7.1) | 5 (9.3) | 1 (3.3) | 0.414+ |
| HbA1c level (%)#^ | 5.65 (5.2–6.5) | 6 (5.5–7.8) | 5.27±0.38 | <0.000* |
| Classification HbA1c | | | | <0.000< |
| HbA1c <5.7%; n (%) | 42 (50) | 17 (31.5) | 25 (83.3) | |
| HbA1c 5.7–6.4%; n (%) | 21 (25) | 16 (29.6) | 5 (16.7) | |
| HbA1c ≥ 6.5%; n (%) | 21 (25) | 21 (38.9) | 0 (0) | |
| FBG level (mg/dL)#^ | 88 (81–111.5) | 97 (85.75–144.25) | 82.87±7.92 | <0.000* |
| FBG ≥100 mg/dL or on therapy; n (%) | 26 (30.9) | 25 (46.3) | 1 (3.3%) | <0.000< |
| TG level (mg/dL)# | 108.5 (76.25–159.75) | 139.5 (100.25–184.25) | 68.5 (55.5–97.25) | <0.000* |

(*Continued*)

**Table 2.** (Continued)

| Variable | Total (n = 84) | MetS (n = 54) | Non-MetS (n = 30) | p-value |
|---|---|---|---|---|
| TG ≥ 150 mg/dL or on therapy; n (%) | 35 (41.7) | 34 (62.9) | 1 (3.3) | <0.000< |
| HDL level (mg/dL)^ | 43.73±9.6 | 40.81±8.65 | 48.97±9.11 | <0.000> |
| HDL< 40 mg/dL (M) or on therapy, n (%) | 23 (27.4) | 20 (37) | 3 (10) | 0.008< |
| HDL < 50 mg/dL (F) or on therapy, n (%) | 34 (40.5) | 29 (53.7) | 5 (16.7) | 0.001< |
| HDL<40 (M) & <50 (F) or on therapy, n (%) | 57 (67.9) | 49 (90.7) | 8 (26.7) | <0.000< |

^Data in mean ± standard deviation;

#Data in median & interquartile range;

<Chi Square,

>T-independent;

*Mann-Whitney;

+Fisher's exact;

MetS: Metabolic syndrome; BMI: Body mass index; BSA: Body surface area; PASI: Psoriasis area severity index; FBG: Fasting blood glucose; TG: Triglycerides; HDL: High density lipoporotein; M: Male; F: Female.

## Association of metabolic syndrome with hard-to-treat psoriasis severity and the number of hard-to-treat area involvement

The prevalence of MetS was 64.3% in this study as shown in Fig 2. Its prevalence was also found significantly higher in the severe hard-to-treat psoriasis patients than in the mild-moderate group. Hard-to-treat psoriasis with PASI score >10 were 3.67 times more likely to have MetS compared to the PASI ≥10 group (78.6% vs. 21.4%, $p$ = 0.006) with an OR of 3.667, 95% CI 1.413–9.514 (Table 3). On the contrary, there were no significant differences in the proportions of MetS either in patients with one (66.7% vs 33.3%), two (54.% vs 45.5%), or more than two (65.7% vs 34.3%) hard-to-treat areas involvement ($p$ = 0.769). The clinical manifestation of hard-to-treat psoriasis shown in Fig 3.

## Hard-to-Treat Psoriasis

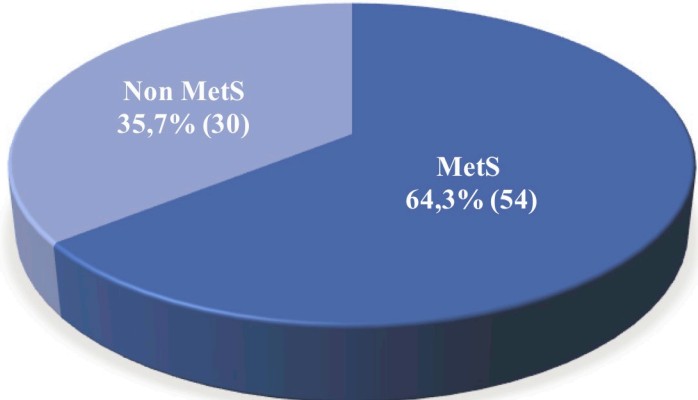

**Fig 2. Prevalence of metabolic syndrome (MetS) in hard-to-treat psoriasis.**

**Table 3. Association of metabolic syndrome and psoriasis severity.**

| Severity (PASI Scores) | MetS (n = 54) | Non-Mets (n = 30) | *p*-value | OR (CI 95%) |
|---|---|---|---|---|
| PASI>10, n (%) | 33 (78.6) | 9 (21.4) | 0.006[<] | 3.667 (1.413–9.514) |
| PASI ≥ 10, n (%) | 21 (50) | 21 (50) | | |

[<]*Chi Square;* OR: Odds ratio; CI: Confidence interval; MetS: Metabolic syndrome; PASI: Psoriasis area severity index.

## Association of metabolic syndrome components and HbA1c level with hard-to-treat psoriasis severity

Abdominal obesity was found to predominate in mild-moderate and severe HTT psoriasis patients (Table 4). The proportion of abdominal obesity was not significantly different between the severe (83.3%) and mild-moderate (81%) hard-to-treat psoriasis groups. On the other hand, severe hard-to-treat psoriasis patients had a 2.67 times higher risk of hypertension than the mild-moderate group (66.7% vs. 42.9%, $p = 0.028$, OR 2.667, 95%; CI 1.099–6.468). The level of FBG in the severe hard-to-treat psoriasis patients was significantly higher compared with the mild-moderate patients (90.5 (84.5–144.26) vs. 84.5 (78.75–98.5) mg/dL, $p = 0.018$). This study showed the prevalence of FBG ≥100 mg/dL in the severe hard-to-treat psoriasis group was 35.7% and in the other group was 26.2% but showed no significant differences.

The TG level in the severe and mild-moderate hard-to-treat psoriasis groups differed significantly (130 (62–141) vs. 95.5 (62–141.5), $p = 0.044$). Contrarily, the prevalence of hypertriglyceridemia in severe hard-to-treat psoriasis was 1.5 higher than in the mild-moderate group but had no significant differences. The mean HDL level comparison between the two groups was not statistically significant in contrast to its proportion. However, patients with severe hard-to-treat psoriasis were 3.5 times more likely to have a low HDL level than the other group (81%

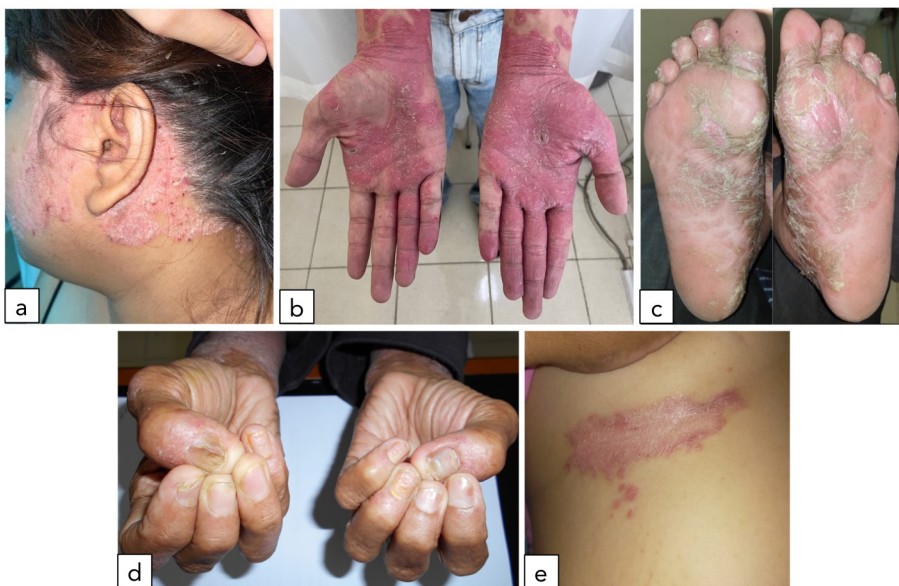

**Fig 3. Psoriatic lesion in some of the hard-to-treat areas of the study population.** Face and scalp (a), palmoplantar (b, c), nails (d), and inframammary folds/intertriginous (e). *Archives of the Dermato-Allergo-Immunology Division, Department of Dermatology & Venereology FMUI/CMH.

**Table 4. Association of metabolic syndrome components and HbA1c level with hard-to-treat psoriasis severity.**

| Variable | PASI > 10 (n = 42) | PASI ≥ 10 (n = 42) | Nilai p |
|---|---|---|---|
| Abdominal obesity; n (%) | 34 (81) | 35 (83.3) | 0.776[<] |
| Hypertension; n (%) | 28 (66.7) | 18 (42,9) | 0.028[<] |
| HbA1c level (%)[#] | 6 (5.17–7.8) | 5.5 (5.2–6) | 0.152[*] |
| FBG level (mg/dL)[#] | 90.5 (84.5–144.26) | 84.5 (78.75–98.5) | 0.018[*] |
| FBG≥100 mg/dL or on therapy, n (%) | 15 (35.7) | 11 (26.2) | 0.345[<] |
| TG level (mg/dL)[#] | 130 (62–141) | 95.5 (62–141.5) | 0.044[*] |
| TG≥150 mg/dL or on therapy, n (%) | 21 (50) | 13 (31) | 0.075[<] |
| HDL level (mg/dL)[^] | 42.17±8.89 | 45.29±10.13 | 0.138[>] |
| HDL<40 (M) & <50 (F) or on therapy, n (%) | 34 (81) | 23 (54.8) | 0.01[<] |
| HDL <40 mg/dL (M) or on therapy, n (%) | 15 (35.7) | 8 (19) | 0.087[<] |
| HDL<50 mg/dL (F) or on therapy, n (%) | 19 (45.2) | 15 (35.7) | 0.374[<] |

[^]Data in mean ± standard deviation;

[#]Data in median & interquartile range;

[<]*Chi Square*;

[>]*T-independent*;

[*]*Mann-Whitney*;

PASI: *Psoriasis area severity index*; FBG: Fasting blood glucose; TG: Triglycerides; HDL: High density lipoporotein; M: Male; F: Female.

vs. 54.8%, *p* = 0.01, OR 3.511 95% CI 1.316–9.364). In this study, comparative analysis of HbA1c levels had no significant differences. There were significant differences in the proportions of hypertension and low HDL level as well as FBG and TG level. The comparative analysis and distribution of TG, HDL, FBG and HbA1c level between each groups shown in Fig 4.

## Discussion

This hospital-based cross-sectional study found a marked increased risk of MetS in individuals with hard-to-treat psoriasis, particularly with a PASI score >10. This study showed a higher prevalence of MetS compared to the global psoriasis and national non-psoriatic population [6,7,13]. Furthermore, every element of MetS was also higher than in previous other studies [2,13,16,17,19]. This is the first research that investigates the relationship between hard-to-treat psoriasis severity and MetS. These results are thought to be related to hard-to-treat involvement. Hard-to-treat areas are a distinct form of psoriasis phenotypes. The appearance of new lesions and distinct phenotypes is more challenging to define [20]. HTT areas are susceptible to the Koebner phenomenon which induces new psoriatic lesions through mast cell signaling pathways involving tryptase IL-6, IL-8, IL-17, IL-36, and other inflammatory mediators [21]. In the HTT area, it is suspected that new lesions are induced by the Koebner phenomenon, and existing lesions persist in the same area. Characteristics of new lesions show plenty of neutrophils, whereas well-established lesions feature a T cell predominance [20]. Neutrophils play an important role in the progression of psoriasis. The development of psoriasis is typically marked by an excessive concentration of neutrophils in both lesions and blood [22]. Following that, it was shown that psoriatic lesions consist of psoriasis-specific tissue-resident memory T lymphocytes which induce IL-17 and IL-22. Moreover, the normal-appearing skin of psoriasis patients also conceives T cells that might generate psoriasis recurrent [23].

The association of MetS and its components (hypertension, abdominal obesity, hyperglycemia, hypertriglyceridemia, and low HDL level) in psoriasis involves genetic factors and

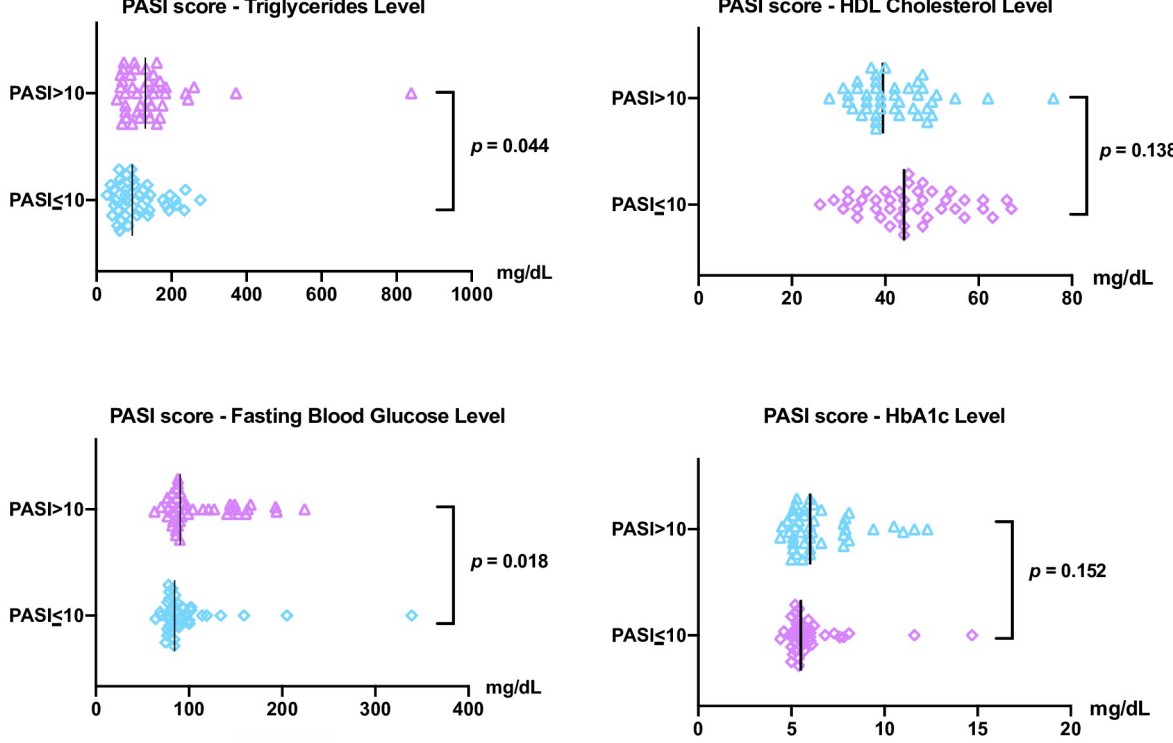

**Fig 4. Comparative analysis and distribution of triglycerides, HDL, fasting blood glucose, and HbA1c level between PASI score more than 10 and below 10.** (Δ, ◊ individuals data from each groups; l median).

overlapping molecular pathways [24]. In psoriasis, abnormal immune cell activation with upregulation of proinflammatory cytokines led to an immunological cascade, oxidative damage, and collagen accumulation in arteries, which may contribute to blood vessel stiffening and hypertension [24,25]. One of the vasoactive peptides that have a key role in vascular homeostasis is endothelin-1. Keratinocytes produce it to constrict blood vessels, increase arterial blood pressure, and maintain vascular tone. Increased endothelin-1 production were as a cause of hypertension in psoriasis [15,24].

Previous studies and this study found no significant differences in the prevalence of abdominal obesity at various levels of psoriasis severity [13,17]. Abdominal obesity is an important element of MetS that presents in most of the subjects. Psoriasis patients were found to have an increased expression of visfatin (a proinflammatory adipokine) in visceral tissues, positively correlated with abdominal obesity [26]. Visfatin mediates complex cellular signaling processes stimulated by oxidative stress that triggers vascular endothelial inflammation, insulin resistance, and dyslipidemia [26,27]. In addition, visfatin can acts on keratinocytes to amplify the inflammatory state through the NF-κB and STAT3 signaling pathways as well as upregulation of several chemokines, thereby increasing the severity of psoriasis [26].

Significantly differences in FBG level indicate a greater risk of insulin resistance in severe HTT psoriasis than mild-moderate. Clinicians must tread cautiously to prevent the emergence of DM from this condition. Several studies have reported that TNF-α, IL-23, IL-17, and adipokines affect insulin sensitivity. These cytokines influenced insulin resistance, by reducing tyrosine-kinase activity on insulin receptors [28]. TNF-α directly contributes to insulin resistance by activating stress kinases, thus blocking insulin signal transduction [1]. In addition, TNF-α

also causes phosphorylation of IRS-1 which triggers decreased expression of GLUT-4, reducing the entry of glucose into cells [24].

Overall proportions of hypertriglyceridemia and low HDL level were found higher compared to other studies [29–31]. Current studies identified HLA genes were associated with psoriasis and positively correlated with the risk of dyslipidemia. Elevated TNF-α, IL-1β, and IL-6 levels in psoriasis reduce triglyceride clearance [24]. Adipose tissue secretes many adipokines, particularly adiponectin which improves insulin resistance and regulates glucose and lipid metabolism. Adiponectin increases HDL and decreases serum TG levels by increasing the catabolism of triglyceride-rich lipoproteins. Serum adiponectin levels are negatively related to TNF-α and IL-6 levels. Increased TNF-α and IL-6 can interfere with adiponectin multimerization, thus decreasing its secretion. Adiponectin levels correlated negatively with PASI score. A decrease in adiponectin at high PASI will cause an increase in triglycerides and a decrease in HDL [1]. Other studies have shown that chronic inflammation can cause changes in structural proteins, including the formation of neo-epitopes, triggering changes in HDL, producing autoantibodies against it. Anti-HDL is detectable in patients with psoriasis, and its presence correlates with disease severity [24,32].

Currently, no similar study that analyzed the association of MetS and the number of HTT lesion areas. The study related to the number of HTT locations reported that from 2042 psoriasis patients, 26.2% had two or more HTT areas [12]. In other diseases, particularly rheumatoid arthritis (AR), the degree of severity determines the development of systemic inflammation. Cardiovascular disease associated with the number of inflamed joints in AR [20]. The results of this study proved that there was no relationship between the number of HTT areas involved and MetS.

The mean age of whole subjects was above 40 years old and even older in the MetS group. MetS prevalence rises with age in both genders due to an increase in chronic illnesses such as hypertension, abdominal obesity, dyslipidemia, diabetes, and sex hormone deficiency as people age [33]. The proportion of smoking and alcohol consumption was low and thus had no significant association with MetS. Patients may have underreported their risks of smoking and drinking due to responder and recall bias [17]. Moreover, alcohol consumption in Indonesia is lower than in other Southeast Asia and Asia Pacific regions. Cultural and religious values strongly influenced this difference [34].

Most patients use methotrexate as systemic therapy (61.9%). This outcome is consistent with Indonesian treatment guidelines that prioritize methotrexate as the first line of psoriasis treatment [35]. Methotrexate is commonly recommended in international psoriasis guidelines due to its cost-effectiveness and safety [36]. Ferdinando et al. reported that methotrexate was the most widely administered systemic treatment in both the SM and non-SM groups [3].

From the several studies above it is known that the most frequently involved HTT area is the scalp [10,11,37]. The scalp is relatively difficult to access by topical treatment [11]. In addition, the scalp area is susceptible to Koebner's phenomenon due to scratching and daily care such as shampooing and combing [38]. Other areas reported with prevalence vary widely between studies, since some researchers combined folds and genitals, and others separate the two areas [10,37]. In addition, some studies only examine part of the HTT area [12].

This study has some limitations. We did not compare it to the control for example a general population or psoriatic patients with no involvement of hard-to-treat areas. As an observational study, there is no interference or manipulation of the research subjects, as well as no control or treatment group. But, we compare the participants into subgroup based on their severity.

## Conclusions

In conclusion, this study indicates that MetS is exceedingly prevalent in hard-to-treat psoriasis. This study also described a possible role of hard-to-treat involvement in the increased risk of MetS and its components in psoriasis The main components of MetS in hard-to-treat psoriasis subsequently were abdominal obesity, decreased levels of high-density lipoprotein, hypertension, hyperglycemia, and hypertriglyceridemia. These findings emphasize the necessity of early detection of cardiometabolic diseases among patients with hard-to-treat psoriasis. More large-scale research is required to determine the pathomechanism of high prevalence of MetS in hard-to-treat psoriasis.

## Supporting information

**S1 File. Psoriasis MetS data.**
(PDF)

## Acknowledgments

The authors would like to express highest gratitude to all psoriasis patients included in this research and the Dermatology and Venereology Outpatient Clinic Dr. Cipto Mangunkusumo National Central General Hospital, Persahabatan Central General Hospital, Tarakan General Hospital, and Sam Marie Wijaya Hospital.

## Author Contributions

**Conceptualization:** Endi Novianto, Windy K. Budianti.

**Data curation:** Dina Evyana, Retno Wibawanti, Saskia A. R. Hapsari, Euis Mutmainnah.

**Formal analysis:** Dina Evyana, Endi Novianto, Windy K. Budianti, Roro I. A. Krisanti, Wismandari Wisnu, Retno Wibawanti, Hanny Nilasari, Lili Legiawati.

**Funding acquisition:** Endi Novianto.

**Investigation:** Dina Evyana, Saskia A. R. Hapsari, Euis Mutmainnah.

**Methodology:** Dina Evyana, Retno Wibawanti.

**Project administration:** Dina Evyana.

**Resources:** Dina Evyana, Saskia A. R. Hapsari, Euis Mutmainnah.

**Software:** Dina Evyana.

**Supervision:** Endi Novianto, Windy K. Budianti, Roro I. A. Krisanti, Wismandari Wisnu, Retno Wibawanti, Hanny Nilasari, Lili Legiawati, Saskia A. R. Hapsari, Euis Mutmainnah.

**Validation:** Endi Novianto, Windy K. Budianti, Roro I. A. Krisanti, Wismandari Wisnu, Retno Wibawanti.

**Visualization:** Dina Evyana.

**Writing – original draft:** Dina Evyana.

**Writing – review & editing:** Endi Novianto, Windy K. Budianti, Roro I. A. Krisanti, Wismandari Wisnu, Retno Wibawanti, Hanny Nilasari, Lili Legiawati.

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
