## [Decision Letter · Decision Letter 0]

22 Oct 2023

PONE-D-23-19694Association between the severity of hard-to-treat psoriasis and the prevalence of metabolic syndrome: A hospital-based cross-sectional study in Jakarta, IndonesiaPLOS ONE

Dear Dr. Novianto,

Thank you for submitting your manuscript to PLOS ONE. After careful consideration, we feel that it has merit but does not fully meet PLOS ONE’s publication criteria as it currently stands. Therefore, we invite you to submit a revised version of the manuscript that addresses the points raised during the review process.

All comments should be sufficiently addressedThis manuscript requires a minor revisionPlease carefully read and address each concern raised by the two reviewers ==============================

We look forward to receiving your revised manuscript.

Kind regards,

Fredirick Lazaro mashili, MD, PhD

Academic Editor

PLOS ONE

Journal Requirements:

4. Please remove your figures from within your manuscript file, leaving only the individual TIFF/EPS image files, uploaded separately. These will be automatically included in the reviewers’ PDF.

Additional Editor Comments:

Please carefully and sufficiently address all the concerns raised by the reviewers

Reviewers' comments:

Reviewer's Responses to Questions

**Comments to the Author**

1. Is the manuscript technically sound, and do the data support the conclusions?

Reviewer #1: Yes

Reviewer #2: Yes

2. Has the statistical analysis been performed appropriately and rigorously? 

Reviewer #1: Yes

Reviewer #2: Yes

3. Have the authors made all data underlying the findings in their manuscript fully available?

Reviewer #1: Yes

Reviewer #2: Yes

4. Is the manuscript presented in an intelligible fashion and written in standard English?

Reviewer #1: Yes

Reviewer #2: Yes

5. Review Comments to the Author

Reviewer #1: General comment.

This study is a replication of numerous earlier, related studies. The writers should succinctly justify the reason for doing it in the Indonesian Population. This can be understood by looking at some of the references in this manuscript, such as references 13, 17, and 19, as well as further uncited work in literature databases. Has ethnicity anything to do with it?

ABSTRACT

Well-written and summarizes important information in the manuscript.

INTRODUCTION

Lines 67 and 68 the statement “To date no studies have assessed the prevalence of MetS in the HTT psoriasis population and its associated factors”. The statement may be overstated and needs to be revised.

METHODS

Give a reference for the classification of patients using PASI scores that you adopted in this study i.e. (PASI scores ≤ 10 and >10).

RESULTS

1. Lines 126 and 127 “Afterwards the participant also divided into PASI<10 (mild=12, moderate=20, total=42) and >10 groups (severe=42)” Revise the sentence.

2. In the entire manuscript, any numbers with decimal places should be corrected. It appears that (, and.) were mixed up when the number was decimalized.

DISCUSSION

Extend the paragraph that describes the study's limitations. Unlike previous writers who employed the case-control design, this study did not use a control group. How did this not compromise the reliability of this study?

Reviewer #2: Abstract

Well summarized and readable and understandable

Introduction

Well narrowed to the aim of the study

Material and methods

Data collection

Line 92-93: why only 1st and 2nd author had access to information that identify the participants. Can the author explain

Line 99-100: How weight and height was measured (the protocol) and what kind of instrument was used? (brand and model). Can the author state

Line 105-106: How was the waist circumference measured, protocol used and how Blood pressure was measured and what machine was used the brand and model. Can the author state.

Results

Line 126-127: there is calculation error of the data of PASI ≤ 10 (mild 12, moderate 20, total 42). The author needs to revise this statement

How did the author take into account the cofounders such as age, life style such as alcohol intake cigarette smoking and physical inactivity as the risk for metabolic syndrome. Can the author explain this?

Discussion

The findings were well discussed in comparison with previous studies and the pathophysiology of the disease development.

Conclusion

Well concluded based on the findings of the study.

6. PLOS authors have the option to publish the peer review history of their article (what does this mean?). If published, this will include your full peer review and any attached files.

Reviewer #1: **Yes: **Oscar Mbembela

Reviewer #2: No

---

## [Author Response · Author response to Decision Letter 0]

13 Feb 2024

Additional requirements.

Author response: Thank you for your reminder. We will ensure our manuscript meets PLOS ONE’s style requirements.

Author response: Thank you for your notification. We have included the screen capture of the Financial Disclosure stated in the manuscript inside the letter labeled "Response to Reviewers". Both statements provide the same grant numbers. We will ensure that we provide the correct grant numbers and information for both sections.

Author response: Thank you for your notification. We have already share the data through the Fairsharing.org as recommended by PLOS One Journal, but still await for the DOI access. Below the screenshot of the fairsharing web page taken on 4th January 2024.

4. Please remove your figures from within your manuscript file, leaving only the individual TIFF/EPS image files, uploaded separately. These will be automatically included in the reviewers’ PDF.

Author response: Thank you for your notification. We will remove all figures from within the manuscript file and upload the individual TIFF/EPS image files separately.

Author response: Thank you for your reminder. We will ensure our reference list is complete and correct

Reviewer's Responses to Questions

Comments to the Author

1. Is the manuscript technically sound, and do the data support the conclusions?

Reviewer #1: Yes

Reviewer #2: Yes

Author Response: Thank you.

2. Has the statistical analysis been performed appropriately and rigorously? 

Reviewer #1: Yes

Reviewer #2: Yes

Author Response: Thank you.

3. Have the authors made all data underlying the findings in their manuscript fully available?

Reviewer #1: Yes

Reviewer #2: Yes

Author Response: Thank you.

4. Is the manuscript presented in an intelligible fashion and written in standard English?

Reviewer #1: Yes

Reviewer #2: Yes

Author Response: Thank you.

5. Review Comments to the Author

Reviewer #1: General comment.

This study is a replication of numerous earlier, related studies. The writers should succinctly justify the reason for doing it in the Indonesian Population. This can be understood by looking at some of the references in this manuscript, such as references 13, 17, and 19, as well as further uncited work in literature databases. Has ethnicity anything to do with it?

Author Response: Thank you for your concern. We have revised the statement to justify the reason for doing the research in the Indonesian Population.

- See line 21-22 … in Indonesian population.

- See line 68 – 75:

Since MetS and psoriasis develop from both genetic and environmental factors, the relationship between psoriasis and MetS may differ amongst different ethnicities or populations due to differences in environmental exposure and genetic background. Although most studies have shown link between psoriasis severity and MetS, similar research among Asian psoriatic patients were limited with varying results. It is not always feasible to generalize the data and findings beyond populations. Therefore, the aim of the present study was to determine whether hard-to-treat psoriasis severity is associated with the increased prevalence of MetS and which components of MetS play an important role in our Indonesian HTT psoriasis population.

ABSTRACT

Well-written and summarizes important information in the manuscript.

Author Response: Thank you

INTRODUCTION

Lines 67 and 68 the statement “To date no studies have assessed the prevalence of MetS in the HTT psoriasis population and its associated factors”. The statement may be overstated and needs to be revised.

Author Response: Thank you for your kind suggestion. It has been revised as requested in the line 67 and 68. 

METHODS

Give a reference for the classification of patients using PASI scores that you adopted in this study i.e. (PASI scores ≤ 10 and >10).

Author Response: Thank you for your kind suggestion. The PASI score classification was adopted from other similar research by Souza et al. (reference number 2), Itani et al. (reference number 13). We have cited the sentences according to the above references (see lines 100). In addition, study of Nada et al. (reference number 29) has the same cut off but different classification. 

RESULTS

1. Lines 126 and 127 “Afterwards the participant also divided into PASI<10 (mild=12, moderate=20, total=42) and >10 groups (severe=42)” Revise the sentence.

2. In the entire manuscript, any numbers with decimal places should be corrected. It appears that (, and.) were mixed up when the number was decimalized.

Author Response: Thank you for your kind suggestion. 

(1) The sentence has been revised as requested in the line 127.

Afterwards the participants were also divided into PASI<10 (mild=12, moderate=20, total=42) and PASI>10 groups (severe=42).

(2) The numbers with decimal places has been corrected. Decimal is using point (.).

DISCUSSION

Extend the paragraph that describes the study's limitations. Unlike previous writers who employed the case-control design, this study did not use a control group. How did this not compromise the reliability of this study?

Author Response: Thank you for your concern. We have extended the paragraph in the line 310 and 314.

Reviewer #2: Abstract

Well summarized and readable and understandable

Author Response: Thank you.

Introduction

Well narrowed to the aim of the study

Author Response: Thank you.

Material and methods

Data collection

Line 92-93: why only 1st and 2nd author had access to information that identify the participants. Can the author explain

Author Response: Thank you for your concern. The 1st and 2nd author had access to all participants’ identity. This is a policy to maintain patient’s confidentiality since the grant is submitted by the first and second author. But the other authors that in charge of treating the patient has an access to his/her patient identity. For example, the author from Tarakan General Hospital have access only to the patient identity from her hospital but not from other hospital. 

Line 99-100: How weight and height was measured (the protocol) and what kind of instrument was used? (brand and model). Can the author state.

Author Response: Thank you for your kind suggestion. We have include the protocol of how the body weight and height was measured, in the page 8-10 of this file. We used the Tesena TSN 9806 WHS medical digital weighing scale and height measuring (stated in the lines 105 – 108). 

Line 105-107: Body weight and height was measured using the Tesena TSN 9806 WHS medical digital weighing scale and height measuring. 

Line 107-108: The weight measurement is recorded in kilograms (kg) and height in meters (m).

Line 105-106: How was the waist circumference measured, protocol used and how Blood pressure was measured and what machine was used the brand and model. Can the author state.

Author Response: Thank you for your kind suggestion. We have added the brief explanation of both measurements in the lines 104-105 and 108-109. We used OneMed® OD-235 waist ruler and Omron® BP 1300 blood pressure monitors. The complete protocol can be found in page 9-10 of this file.

Line 104-105: Using a flexible tape OneMed® OD-235, the waist circumference was measured midway between the iliac crest and the lowest rib. 

Line 108-109: Blood pressure was measured using Omron® BP-1300 blood pressure monitors.

Results

Line 126-127: there is calculation error of the data of PASI ≤ 10 (mild 12, moderate 20, total 42). The author needs to revise this statement

Author Response: Thank you for your kind notification. We have revised the statement in the line 136: …….PASI<10 (mild=12, moderate=30, total=42)……

How did the author take into account the cofounders such as age, life style such as alcohol intake cigarette smoking and physical inactivity as the risk for metabolic syndrome. Can the author explain this?

Author Response: Thank you for your kind suggestion. We did multivariate analysis about the confounding factors mention above. In this study, alcohol, and smoking were not confounders. The independence factors associated with MetS are age, severity, and body mass index with Nagelkerke R square 0.553. 

Discussion

The findings were well discussed in comparison with previous studies and the pathophysiology of the disease development.

Author Response: Thank you.

Conclusion

Well concluded based on the findings of the study.

Author Response: Thank you.

6. PLOS authors have the option to publish the peer review history of their article (what does this mean?). If published, this will include your full peer review and any attached files.

Author Response: Thank you for the option. Yes, we agree to include it. 

Do you want your identity to be public for this peer review? For information about this choice, including consent withdrawal, please see our Privacy Policy.

Reviewer #1: Yes: Oscar Mbembela

Reviewer #2: No

Author Response: Thank you to both reviewers

Protocol Weight Measurement

1. Weight is measured with a calibrated, electronic digital scale. We use Tesena® TSN 9806 WHS medical digital weighing scale and height measuring.

2. The scale should be placed on a hard-floor surface (not on a floor which is carpeted or otherwise covered with soft material). 

3. Identify the subject and explain the procedure. 

4. The weight measurement is recorded in kilograms (kg).

5. Place a clean paper towel on the scale foot stand. Zero the scale. 

6. The digital LED readout should show 000.00 before weighing a sampled person. If it does not, press the zero key on the scale to zero. 

7. Participants are asked to remove their heavy outer garments (jacket, coat, trousers, skirts, etc.), shoes, and any other articles of clothing or jewelry. Participants should be in minimal underclothing. They are asked to wear an examination gown (light clothes) and to remove objects such as cell phones or wallets from their hands or pockets. 

8. Participants should be weighed at the same time of the day, if possible, after voiding. 

9. The health provider directs participants to stand in the center of the scale platform with hands at their sides and looking straight ahead. They should stand still with weight evenly distributed on both feet. 

10. Read and record the weight accurately to the resolution of the scale (the nearest 0.1 kg)

11. Calibration should occur at the beginning and end of each examining day. For calibrating an electronic scale, follow the instructions of the specific scale. Note that the reading of an electronic scale depends on the gravity of each location. Therefore its calibration is particularly important whenever a new examination site is set up. 

Protocol Height Measurement

1. The height was measured using the Tesena® TSN 9806 WHS medical digital weighing scale and height measuring. The height rule is taped vertically to the hard flat wall surface with the base at floor level. The floor surface next to the height rule must be hard.

2. The participants asked to remove hair ornaments, jewelry, buns, braids from the top of the head, and shoes. 

3. The participant is asked to stand with his/her back to the height rule. The back of the head, back, buttocks, calves and heels should be touching the stadiometer, feet together. The top of the external auditory meatus (ear canal) should be level with the inferior margin of the bony orbit (cheek bone). The participant is asked to look straight. 

4. The head piece of the stadiometer or the sliding part of the measuring rod is lowered so that the hair (if present) is pressed flat. 

5. Height is recorded to the resolution of the height rule (i.e. nearest millimeter/half a centimeter). If the participant is taller than the measurer, the measurer should stand on a platform so that he/she can properly read the height rule. 

6. If the person is taller then the maximum height of the stadiometer, the self reported height is acceptable and recorded on the collection form. 

7. Calibration occur at the beginning and end of each examination day, the height rule should be checked with standardized rods and corrected if the error is greater than 2 mm.

Protocol Blood Pressure Measurement

Conditions 

• Quiet room with comfortable temperature. 

• Before measurements: Avoid smoking, caffeine and exercise for 30 min; empty bladder; remain seated and relaxed for 3–5 min. 

• Neither patient nor staff should talk before, during and between measurements. 

Positions 

• Sitting: Arm bare and resting on table with mid-arm at heart level; back supported on chair; legs uncrossed and feet flat on floor (Figure 1 in the file labeled "Response to Reviewers). 

Device 

• Validated electronic (oscillometric) upper-arm cuff device. Lists of accurate electronic devices for office, home and ambulatory BP measurement in adults, children and pregnant women are available at www.stridebp.org. (We used Omron HBP-1300)

• Alternatively use a calibrated auscultatory device, (aneroid, or hybrid as mercury sphygmomanometers are banned in most coun

---

## [Decision Letter · Decision Letter 1]

3 Apr 2024

Association between the severity of hard-to-treat psoriasis and the prevalence of metabolic syndrome: A hospital-based cross-sectional study in Jakarta, Indonesia

PONE-D-23-19694R1

Dear Dr. Novianto,

We’re pleased to inform you that your manuscript has been judged scientifically suitable for publication and will be formally accepted for publication once it meets all outstanding technical requirements.

Kind regards,

Fredirick Lazaro mashili, MD, PhD

Academic Editor

PLOS ONE

Additional Editor Comments (optional):

You have sufficiently addressed all the previously raised comments

Reviewers' comments:

Reviewer's Responses to Questions

**Comments to the Author**

1. If the authors have adequately addressed your comments raised in a previous round of review and you feel that this manuscript is now acceptable for publication, you may indicate that here to bypass the “Comments to the Author” section, enter your conflict of interest statement in the “Confidential to Editor” section, and submit your "Accept" recommendation.

Reviewer #1: All comments have been addressed

Reviewer #3: All comments have been addressed

2. Is the manuscript technically sound, and do the data support the conclusions?

Reviewer #1: Yes

Reviewer #3: Yes

3. Has the statistical analysis been performed appropriately and rigorously? 

Reviewer #1: Yes

Reviewer #3: Yes

4. Have the authors made all data underlying the findings in their manuscript fully available?

Reviewer #1: (No Response)

Reviewer #3: Yes

5. Is the manuscript presented in an intelligible fashion and written in standard English?

Reviewer #1: Yes

Reviewer #3: Yes

6. Review Comments to the Author

Reviewer #1: All comments have been adequately addressed and therefore at the editorial office's discretion this manuscript may be accepted for publication

Reviewer #3: All the previously raised comments have been thoroughly and sufficiently addressed. The authors have considered and adhered to journal's style and requirements.

7. PLOS authors have the option to publish the peer review history of their article (what does this mean?). If published, this will include your full peer review and any attached files.

Reviewer #1: **Yes: **Oscar Mbembela

Reviewer #3: **Yes: **Fredirick Mashili
